METHODS AND RESOURCES

# Seasonal tissue-specific gene expression in wild crown-of-thorns starfish reveals reproductive and stress-related transcriptional systems

**Marie Morin**[1☯], **Mathias Jönsson**[1☯], **Conan K. Wang**[2], **David J. Craik**[2], **Sandie M. Degnan**[1]*, **Bernard M. Degnan**[1]*

**1** Centre for Marine Science, School of the Environment, The University of Queensland, Brisbane, Australia,
**2** Institute for Molecular Bioscience, ARC Centre of Excellence for Innovations in Peptide and Protein Science, The University of Queensland, Brisbane, Australia

☯ These authors contributed equally to this work.
* s.degnan@uq.edu.au (SMD); b.degnan@uq.edu.au (BMD)

**Data Availability Statement:** All relevant data and scripts are within the paper and Supporting Information and in the Zenodo public repository

## Abstract

Animals are influenced by the season, yet we know little about the changes that occur in most species throughout the year. This is particularly true in tropical marine animals that experience relatively small annual temperature and daylight changes. Like many coral reef inhabitants, the crown-of-thorns starfish (COTS), well known as a notorious consumer of corals and destroyer of coral reefs, reproduces exclusively in the summer. By comparing gene expression in 7 somatic tissues procured from wild COTS sampled on the Great Barrier Reef, we identified more than 2,000 protein-coding genes that change significantly between summer and winter. COTS genes that appear to mediate conspecific communication, including both signalling factors released into the surrounding sea water and cell surface receptors, are up-regulated in external secretory and sensory tissues in the summer, often in a sex-specific manner. Sexually dimorphic gene expression appears to be underpinned by sex- and season-specific transcription factors (TFs) and gene regulatory programs. There are over 100 TFs that are seasonally expressed, 87% of which are significantly up-regulated in the summer. Six nuclear receptors are up-regulated in all tissues in the summer, suggesting that systemic seasonal changes are hormonally controlled, as in vertebrates. Unexpectedly, there is a suite of stress-related chaperone proteins and TFs, including HIFa, ATF3, C/EBP, CREB, and NF-κB, that are uniquely and widely co-expressed in gravid females. The up-regulation of these stress proteins in the summer suggests the demands of oogenesis in this highly fecund starfish affects protein stability and turnover in somatic cells. Together, these circannual changes in gene expression provide novel insights into seasonal changes in this coral reef pest and have the potential to identify vulnerabilities for targeted biocontrol.

(https://doi.org/10.5281/zenodo.10831187). The raw RNA sequences used in this study are publicly available in NCBI Sequence Read Archive (SRA) under BioProject PRJNA821257. New GBR v1.1 gene models and transcriptomes generated can be visualized at: https://apollo-portal.genome.edu.au/degnan/cots/.

**Funding:** This research was supported by a Linkage Project grant (LP170101049) from the Australian Research Council to BMD, DJC, SMD and CKW. DJC and CKW are supported by a Fellowship from the National Health and Medical Research Council, Australia (2009564) and by access to the facilities of the Australian Research Council Centre of Excellence for Innovations in Peptide and Protein Science (CE200100012) and an ARC Future Fellowship (FFT220100583). The funders had no role in study design, data collection and analysis, decision to publish, or preparation of the manuscript.

**Competing interests:** The authors have declared that no competing interests exist.

**Abbreviations:** COTS, crown-of-thorns starfish; FDR, false discovery rate; GO, Gene Ontology; GPCR, G-protein coupled receptor; HSP, heat shock protein; PCA, principal component analysis; TF, transcription factor; TPM, transcripts per million; VST, variance-stabilising transformed; WGCNA, weighted gene correlation network analysis.

## Introduction

The crown-of-thorns starfish (COTS, *Acanthaster "planci"* species complex) is a predator of reef-building corals that lives in the oceans of the Indo-Pacific [1–5]. The very high seasonal fecundity of this starfish, along with the timing of its spawning and the high dispersal and survival potential of its larvae, appear to contribute to population fluctuations and outbreaks that cause extensive loss of coral cover and associated reef biodiversity [4,6–16].

COTS, as is the case with many echinoderms and other marine invertebrates, reproduce in the summer by broadcast spawning gametes into the water column where fertilization occurs. Prior to this spawning event, individual starfish come together in response to conspecific and environmental cues to form aggregations of multiple individuals [8,10,11,17–20]. Understanding the molecular basis of the conspecific communication required for aggregation and spawning could provide novel ways to disrupt the starfish reproduction and thus mitigate destructive outbreaks [21]. As gene expression levels can provide insights into the regulatory and physiological state of an organism [22–27], transcriptomic analyses have the potential to identify genes underlying the mechanisms used by COTS to adjust their physiology and behaviours according to time of year. For example, gravid males and females differentially express both genes encoding signalling factors that are released into the surrounding sea water and chemosensory receptors that potentially detect these signals [21,28].

To characterise how COTS change seasonally, we assessed gene expression in 7 somatic tissues isolated immediately after the starfish were hand-collected on the Great Barrier Reef in the summer and winter. This sampling strategy allowed us to identify genes that are differentially expressed between gravid, aggregating female and male starfish prior to summer spawning. To the best of our knowledge, this study provides the first high-resolution, tissue-specific seasonal transcriptomes for any marine animal. Indeed, the first transcriptomes to reveal how human tissues change according to major cycling environmental conditions were published only in 2023 [26].

## Results

### Sampling wild crown-of-thorns starfish on the Great Barrier Reef

We analysed CEL-Seq2 transcriptomes generated from RNA procured from concentrated coelomocytes, and small, targeted biopsies of papulae, radial nerve, sensory tentacles, skin, spines, and tube feet from 20 individual COTS removed by hand from Davies and Lynch's Reefs on the Great Barrier Reef (Fig 1A and S1 Table; see Materials and methods and [28] for details). Seven females and 6 males were sampled in the summer, and 7 unsexed individuals in the winter (these animals could not be sexed because no gonads were visible). All COTS collected in the summer were gravid, indicating that a spawning event was imminent [8,17,28]. Consistent with this inference, some individuals began spawning upon collection. All individuals were maintained in ambient seawater on a vessel anchored at the collection site until dissection and were dissected within 2 h of removal from the reef. At the times of field collection, the mean summer and winter sea surface temperatures were 27.7 and 23.0˚C, respectively, which are within the normal seasonal range for the region (Fig 1B). Day lengths, as measured from sunrise to sunset, were approximately 13.2 and 11.3 hours in the summer and winter, respectively. There were no exceptional weather events prior to, or at the time of, sampling.

Although we procured small biopsies from specific regions of the external tissues (e.g., single tube foot per individual) and restricted the coelomic fluid to a few ml (see Materials and methods), we acknowledge that the transcriptomes nonetheless are likely to be derived from multiple cell types. However, for simplicity we refer to these as "tissues" throughout this report.

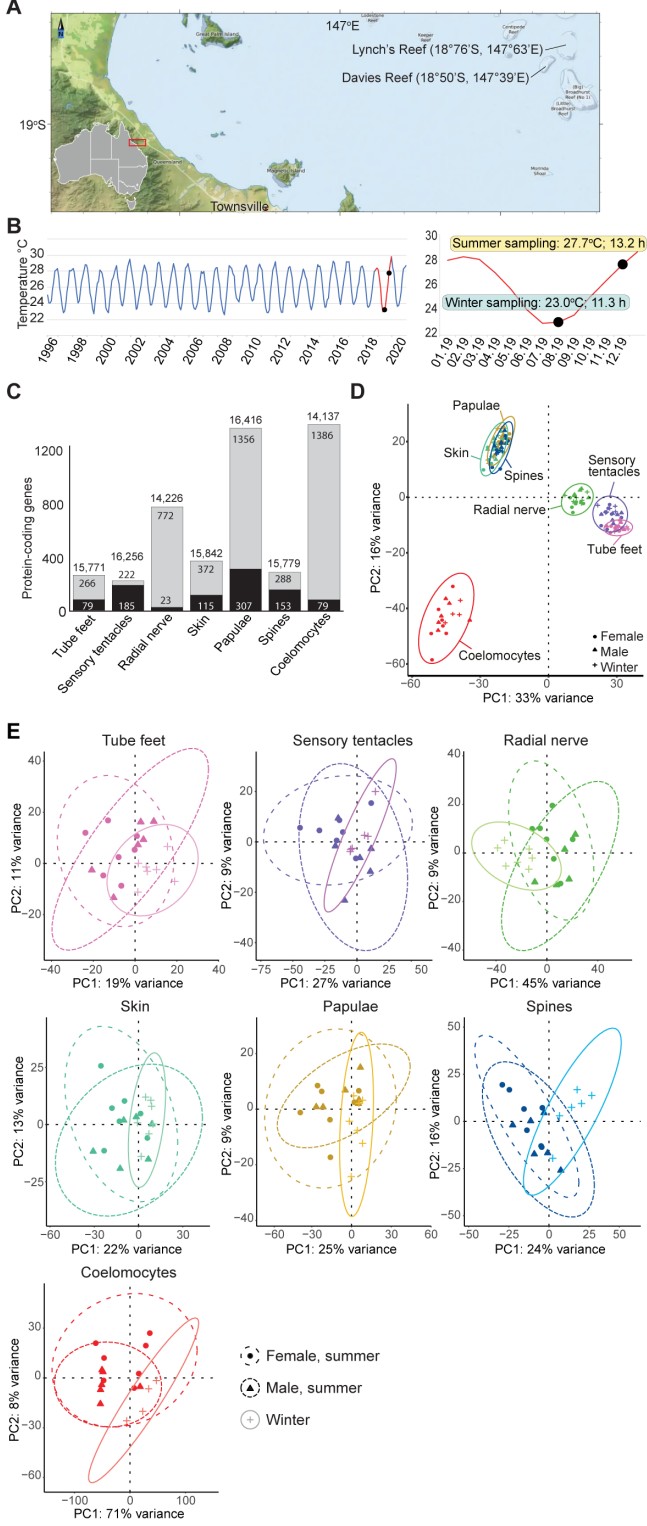

**Fig 1. Gene expression in wild COTS.** **(A)** Locations of field sites—Davies and Lynch's Reefs on the Great Barrier Reef—where RNA was procured from COTS tissues in 2019. **(B)** Sea temperatures at 4 m depth at Davies Reef from 1996 to 2020, and 2019 is highlighted in red and the months of COTS sampling in that year are marked with black dots. Water temperature and day length (sunrise to sunset) at sampling times are shown. Map and temperature data were obtained from the Australian Institute of Marine Science (https://www.aims.gov.au/reef-monitoring/townsville-sector-2017 and https://apps.aims.gov.au/metadata/view/0ba80e4d-eeb9-4b34-9fff-fc6d2787bad1). **(C)** The number of

protein-coding genes significantly up-regulated in each tissue compared to all other tissues (DESeq2, adjusted *p*-value <0.05). The grey bars and corresponding values are the number of up-regulated genes in each tissue. The black bars and corresponding values are the number of protein-coding genes expressed exclusively in that tissue. The values above the bars are the total number of protein-coding genes expressed in each tissue. **(D)** PCA with 95% confidence ellipses, showing the differences in overall gene expression between each of the sampled tissues. **(E)** PCAs with 95% confidence ellipses, showing the differences in overall gene expression between males and females (summer), and unsexed COTS (winter). Pink, tube feet; purple, sensory tentacles; green, radial nerve; turquoise, skin; brown, papulae; blue, spines; red, coelomocytes; circles, females; triangles, males; crosses, unsexed winter individuals. The gene expression data underlying the PCAs (**D**, **E**) can be found in S1 and S2 Tables and at https://doi.org/10.5281/zenodo.10831187 and can be visualised at https://apollo-portal.genome.edu.au/degnan/cots/ [28,29]. RNA sequence data are available in the NCBI Sequence Read Archive under BioProject PRJNA821257. COTS, crown-of-thorns starfish; PCA, principal component analysis.

## Somatic tissue gene expression

On average, 69.7% of the CEL-Seq2 reads from the 134 tissue transcriptomes that passed quality control filtering mapped to the GBR v1.1 gene models (see Materials and methods) [29], with each tissue expressing on average 15,490 protein-coding genes (Fig 1C and S1 and S2 Tables). Principal component analysis (PCA) and hierarchical clustering analysis revealed that gene expression in each tissue is very similar between individuals, regardless of sex or season (Figs 1D and S1). This affirmed that our sampling regime yielded consistent tissue-specific gene expression profiles.

   The transcriptomes of oral (downwardly-facing) radial nerve, sensory tentacles, and tube feet are most similar to each other and are distinct from those of the aboral (upwardly-facing) papulae, skin, and spines, which are also most similar to each other (Figs 1D and S1). The coelomocytes, the only internal tissue analysed, are distinct from both oral and aboral tissues. Pairwise comparisons of gene expression using DESeq2 identified between 222 and 1,386 protein-coding genes that are significantly up-regulated within a given tissue compared to all other tissues (adjusted *p*-value <0.05; Fig 1C and S3 Table). The 3 oral tissues together uniquely up-regulate 1,752 genes that are enriched in neural and sensory functions, consistent with the known biological roles of these tissues [30–34]; 71.9% of these genes are up-regulated in only 1 tissue (S4 Table). In contrast, 2,786 protein-coding genes up-regulated in aboral tissues are enriched in immune and cilia function, again consistent with known roles of these tissues [35,36]; 72.3% of these are uniquely up-regulated in only 1 tissue. Nearly 1,400 protein-coding genes are up-regulated in the internal coelomocytes compared to the 6 external tissues (S3 Table).

## Seasonal changes in gene expression

Like most marine animals, COTS are poikilotherms (ectotherms) and thus their metabolism and physiology are strongly influenced by the surrounding water temperature [37,38]. COTS inhabiting Davies and Lynch's Reefs in the central part of the Great Barrier Reef typically experience a 6 to 7°C annual change in water temperature and a maximum of 2.3 h difference in day length (sunrise to sunset on summer and winter solstice). These environmental variations are sufficient for COTS and a diversity of coral reef animals to seasonally change their reproductive status, with COTS typically spawning in the austral summer from November to January [8,10,17,19]. The exact timing of spawning appears to be based on a combination of abiotic and biotic cues [10,17,18,20,21,28]. At the time that we undertook the summer sampling (December 2019), all COTS were gravid and appeared to have not yet spawned, consistent with previous observations of the reproductive status of this starfish on Davies Reef [17]. In contrast, gonads were not visible in any of the individuals sampled in the winter (August 2019)

and thus their sex or possible level of hermaphrodism [39] could not be determined. In any case, winter transcriptomes clustered by tissue type with male and female summer transcriptomes, indicating that tissue type was the greatest determinant of transcriptional state in both seasons, regardless of sex (Fig 1D and 1E).

We identified 2,079 protein-coding genes significantly differentially expressed between seasons in at least 1 tissue (adjusted $p$-value $<0.05$); 71.8% of these were up-regulated in summer (Figs 2A and S1 and S5 Table). All tissues follow this general pattern of seasonal gene expression except for the skin, which has the smallest seasonal difference and often has the opposite gene expression profile in the summer to all other tissues (Figs 2A and S1 and S5 Table).

Protein processing pathways were more activated in most tissues in summer, consistent with the up-regulation of multiple protein chaperones, cell surface receptors (including 29 G-protein coupled receptors, GPCRs), and secreted proteins that include conserved hormones, neuropeptides, and developmental proteins (Fig 2B and 2C and S5 Table). Notably, the external sensory tissues—radial nerve, spines, and tube feet—had the greatest seasonal differences in gene expression (Figs 2A–2C and S1 and S5 Table). In the radial nerve, this includes the up-regulation in the summer of components of Wnt, FoxO, AGE-RAGE, and phosphatidylinositol signalling pathways, consistent with a change in physiological and cell states, and an increase in overall intercellular signalling (Fig 2B and 2C).

We previously found that exoproteomes released into the sea water by aggregating COTS are enriched in lineage-specific sequences, such as COTS-specific ependymins [21]. In the present study, we found 285 lineage-specific protein-coding genes that are differentially expressed between summer and winter (S2 Fig and S7 Table). Of these, 202 are up-regulated in the summer, including 16 genes that encode proteins secreted by aggregating COTS and 7 genes that are expressed in a sex-specific manner [21,28]. Notably, we found that spines differentially express the largest number of lineage-specific genes (155), consistent with previous observations that spines and skin release many COTS-specific proteins into the surrounding sea water when COTS are aggregating or when they are alarmed by the presence of a predator [21].

## Sex-specific gene regulation in the summer

We previously identified 183 and 100 protein-coding genes that are differentially up-regulated respectively in female and male somatic tissues just prior to summer spawning [28]. Here, we found evidence that these genes are seasonally regulated in a sex-specific manner, with relative winter transcript levels intermediate between those of the 2 sexes in the summer (Fig 2D and S6 Table). These gene expression profiles are consistent with sex-specific changes in physiological state as environmental cues, including increasing water temperature and day length, trigger a preparation for reproduction. Most of these genes, which appear to be related to the onset of reproduction, are expressed in the sensory tentacles and spines, both of which are implicated in conspecific communication during summer aggregation and in the regulation of spawning [28]. We also detected 21 class A rhodopsin-like GPCRs (putative chemoreceptors) and 189 secreted proteins that are differentially up-regulated in the summer (S2 Fig and S7 Table) [28]. Together, these findings reveal a starfish-wide sex-specific up-regulation of gene expression in the summer that appears to underlie conspecific communication and reproduction.

## Activation of transcription factors in the summer

As part of the general up-regulation of genes in the summer, we found that 89 of 102 seasonally expressed TFs are significantly up-regulated in summer (adjusted $p$-value $<0.05$; Figs 2E, 2F,

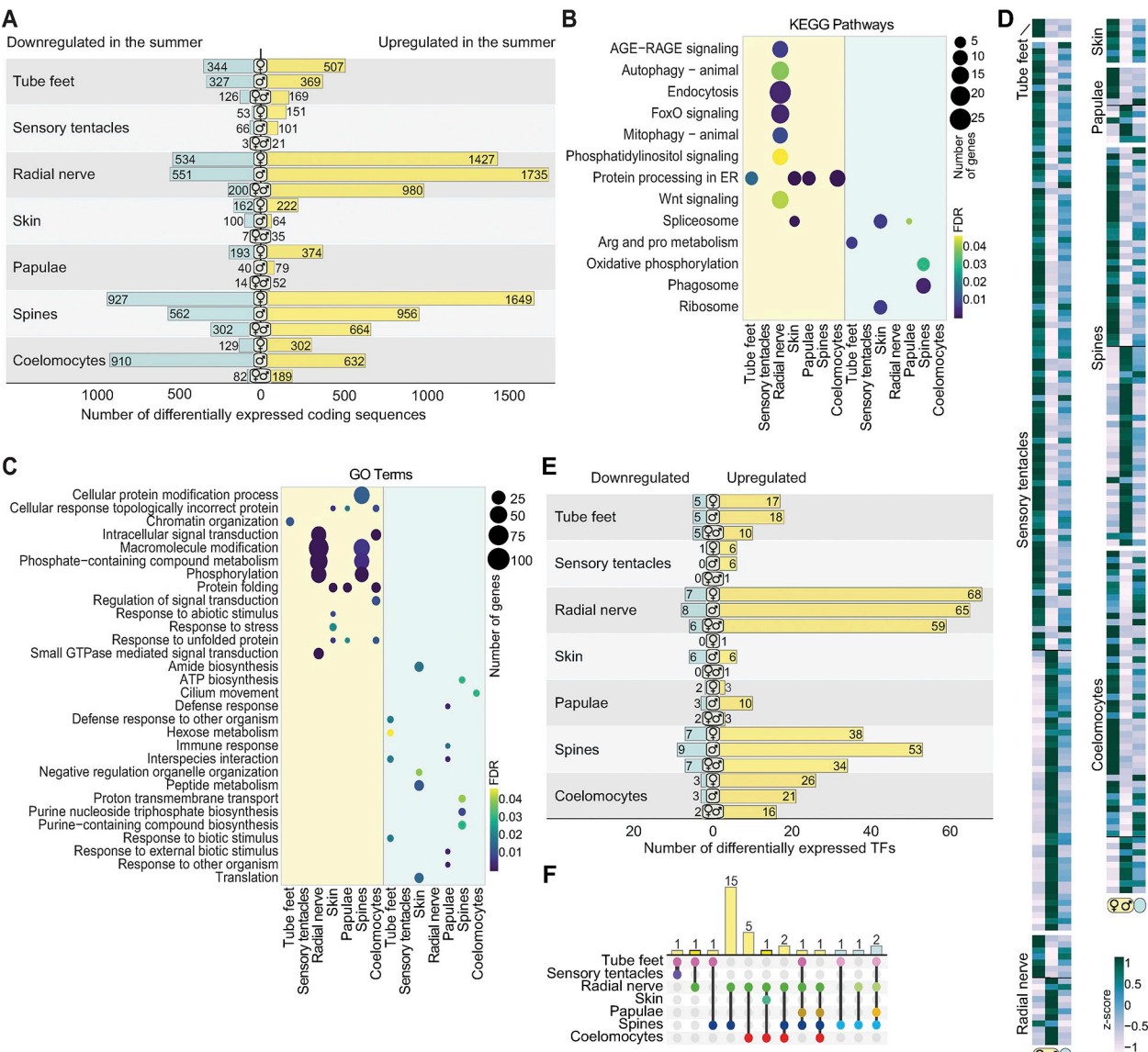

**Fig 2. Differential gene expression between seasons and sexes. (A)** The number of significantly up-regulated (yellow) and down-regulated (blue) protein-coding genes in individual COTS tissues in the summer compared to unsexed winter individuals (adjusted *p*-value <0.05); differentially expressed genes in females (♀), males (♂), or in both sexes (♀♂) are shown (S5 Table). **(B)** KEGG pathways and **(C)** GO terms enriched in the summer (yellow) and winter (blue) based on significantly differentially expressed protein-coding genes in each tissue. The dot size and colour corresponds to the number of protein-coding genes associated with each term and the FDR-corrected *p*-value, respectively (S5 Table). **(D)** Comparison of genes that are significantly differentially expressed between gravid female and male COTS prior to spawning (adjusted *p*-value <0.05) [28] with their expression in the winter. The heatmap shows scaled (z-score) expression levels based on TPM normalised reads from 7 gravid females (♀) and 6 gravid males (♂) in the summer (yellow oval), and 7 unsexed individuals in the winter (blue oval). The heatmap is grouped into the tissues where these genes are differentially expressed (S6 Table). **(E)** The number of significantly up-regulated (yellow) and down-regulated (blue) TF genes in individual COTS tissues in the summer compared to unsexed winter individuals (adjusted *p*-value <0.05); differentially expressed genes in females (♀), males (♂), or in both sexes (♀♂) shown (S5 Table). **(F)** The number of significantly up-regulated TFs expressed in multiple tissues in the summer (left, yellow) and winter (right, blue). The expressed TFs shared by more than 1 tissue are indicated at the bottom and the number of shared TFs by specific tissue combinations are above the bars. The data underlying this figure can be found in S5 and S6 Tables and at [https://doi.org/10.5281/zenodo.10831187](https://doi.org/10.5281/zenodo.10831187). COTS, crown-of-thorns starfish; FDR, false discovery rate; GO, Gene Ontology; TF, transcription factor; TPM, transcripts per million.

[https://doi.org/10.1371/journal.pbio.3002620.g002](https://doi.org/10.1371/journal.pbio.3002620.g002)

and S1 and S5 Table). Although most TFs have unique expression profiles across tissues, sexes, and seasons (61 are uniquely up-regulated in 1 specific tissue), the overall expression profile aligns with sex and season. That is, TF expression is higher in either males or females in summer, or is higher in 1 season compared to the other (Figs 2E, 2F, and S1). We did not find any significantly up-regulated TFs that are shared among all oral tissues or all aboral tissues (Fig 2E and 2F), despite the overall transcriptomic similarities of the tissues comprising these 2 tissue sets (Fig 1C and 1D). These findings suggest that multiple cell type-specific gene regulatory networks are operating across the somatic cells comprising these tissues. Interestingly, seasonal expression of most TFs in the skin is opposite to other tissues by being relatively higher in the winter (S1 Fig). This TF profile is similar to the overall seasonal differential gene expression profile in the skin, which differs markedly from all other tissues (S1 Fig and S5 Table).

Of the 61 TFs that are significantly up-regulated in 1 specific tissue in the summer, 33 and 14 are uniquely up-regulated in radial nerves and spines, respectively (Fig 2E and 2F and S5 Table). These 2 functionally different and oppositely facing tissues also have the most TFs in common (15), including 3 nuclear receptors: nuclear receptor subfamily 1 group D member 1 (NR1D1/Rev-Erbα); nuclear receptor subfamily 1 group D member 2 (NR1D2/ Rev-Erbβ); and peroxisome proliferator-activated receptor gamma (PPARγ) (S1 Fig and S5 Table). The scale and complexity of TF gene expression in spines and the radial nerve suggests that multiple seasonal gene regulatory networks are operating simultaneously across the various cell types that comprise these 2 somatic tissues. Consistent with this suggestion, these 2 tissues also have the most differentially expressed protein-coding genes (Fig 2A).

## Co-expressed summer genes suggest an overall increase in somatic cellular activity and turnover in gravid crown-of-thorns starfish

A weighted gene correlation network analysis (WGCNA) on 9,159 protein-coding genes that met expression and variance thresholds (see Materials and methods) [40] identified 15 gene co-expression modules, most of which are correlated with a specific tissue (S3 Fig and S8 Table). However, 3 of the co-expression modules—turquoise, cyan, and pink—comprise genes that were highly expressed in all tissues in the summer (p-values $1e^{-3}$, $4e^{-2}$, and $4e^{-6}$, respectively; Figs 3A and S3 and S8 Table). There were no co-expression modules that comprise genes up-regulated in all tissues in the winter.

The most interconnected protein-coding genes in the turquoise and cyan modules have similar expression in both male and female COTS and are enriched in genes involved in gene regulation, signal transduction, and enzyme function (Figs 3B and S3 and S8 Table). These expression profiles are consistent with an overall increase in cellular activity and cell turnover in summer compared to winter. The most interconnected gene in this network is a member of the SPEN family of transcriptional repressors, which can interact with nuclear receptors and other corepressors, and which contains RNA recognition motifs that can interact with nuclear receptor mRNAs [41,42]. The turquoise module also includes 6 nuclear receptors—retinoid X receptor (RXR), vitamin D3 receptor B (VDR), ecdysone receptor (EcR/N1NH2), NR1D1/ Rev-Erbα, NR1D2/ Rev-Erbβ, and PPARγ—that may be SPEN targets in somatic tissues of gravid COTS (S3 Fig and S5 and S8 Tables). These nuclear receptors generally are activated by diverse steroid ligands [43], suggesting that summer gene expression profiles are underpinned by a seasonal change in composition of the endogenous ligands (e.g., hormones) to these, and perhaps other constitutively expressed nuclear receptors.

Notably, like the general seasonal TF gene expression profiles in COTS (S1 Fig), all the most interconnected co-expressed up-regulated protein-coding genes in the summer are down-regulated in the skin (S1 and S3 Figs and S5 and S8 Tables). This finding suggests that

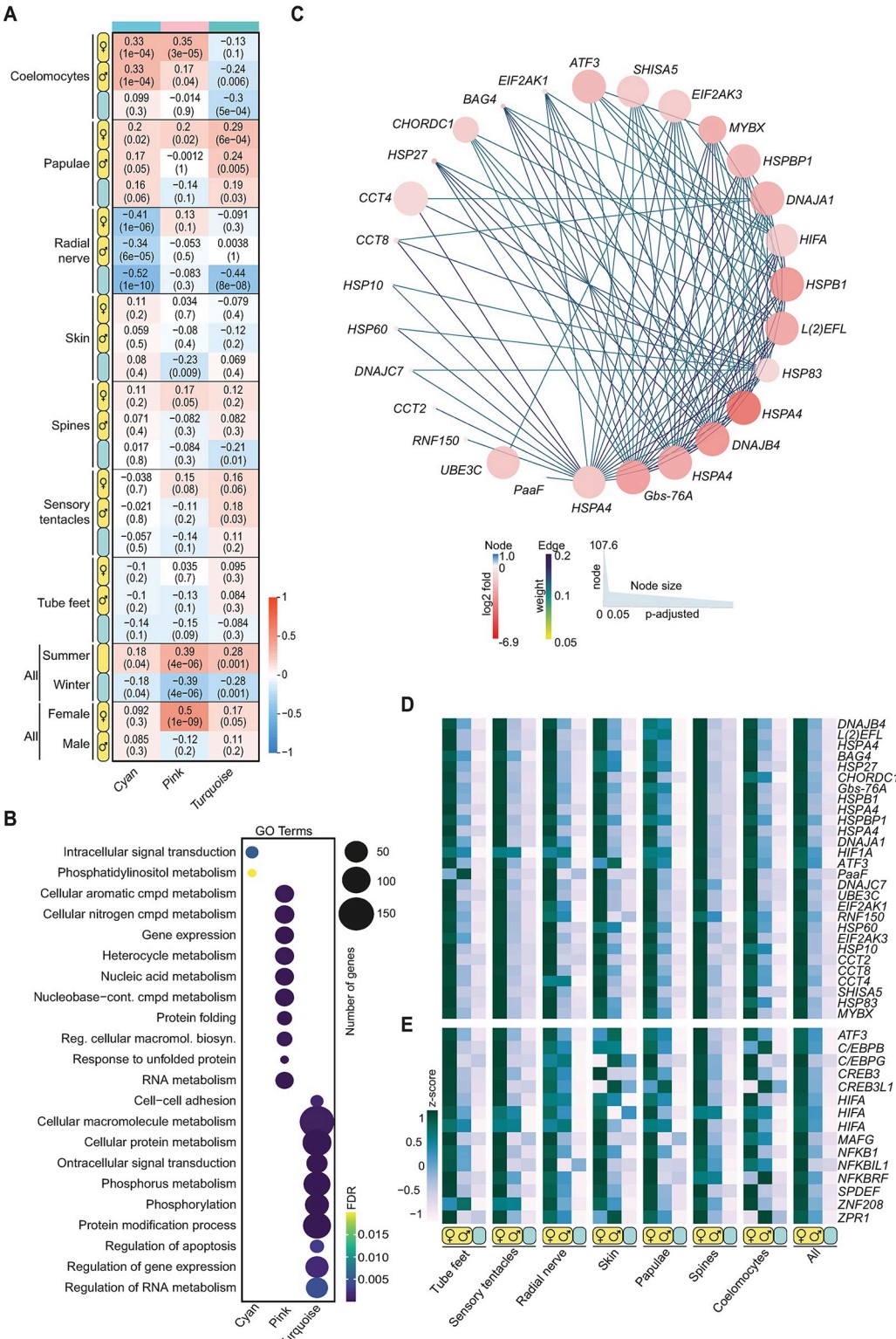

**Fig 3. Seasonal gene co-expression. (A)** Correlation coefficient and *p*-value (in brackets) of co-expressed protein-coding genes in 3 modules (cyan, pink, and turquoise) comprising genes that are up-regulated in the summer in relation to tissue, season, and sex (S8 Table). Red and blue colours depict the strength of the correlation. **(B)** Most significantly enriched biological processes GO terms of the protein-coding genes comprising cyan, pink, and turquoise modules. The dot size reflects the number of genes in each GO term and the colour depicts the FDR-corrected *p*-value. **(C)** Gene

interaction networks for the pink (up-regulated in gravid females) module (edge weights >0.164). The size of each protein-coding gene (node) in the network corresponds to the significance of differential expression (up-regulated in the summer; *p*-adjusted value in DESeq2). The colour corresponds to the log2fold change in gene expression: red, up-regulated in summer; blue, down-regulated in summer. Expression profiles of the core hub genes **(D)** and TFs **(E)** identified in the pink module. These heatmaps show the scaled expression levels (z-score) based on TPM normalized reads for each tissue in females, males, and winter individuals. The "All" heatmap shows the average expression across all groups and tissues. The data underlying this figure can be found in S8 Table and at https://doi.org/10.5281/zenodo.10831187. FDR, false discovery rate; GO, Gene Ontology; TF, transcription factor; TPM, transcripts per million.

this protective tissue has a markedly different regulatory architecture to the other tissues. Skin shares only 1 differentially expressed TF gene with the radial nerve and coelomocytes (*ATF3*) and has no uniquely expressed TFs that are significantly differentially expressed between seasons (Figs 2E, 2F, and S1 and S5 Table). These gene expression and co-expression profiles provide further evidence that of all the tissues skin changes the least in function between winter and summer, consistent with a year-round role in immunity and defense.

## Female crown-of-thorns starfish up-regulate stress proteins in the summer

The pink module (see above) comprises protein-coding genes that are up-regulated in gravid female tissues in the summer (*p*-value 1e$^{-9}$; Fig 3A and S8 Table). This sex- and season-specific co-expression module is enriched in genes related to protein turnover and stability, and other stress-related functions. It includes heat shock proteins (HSPs), hypoxia-inducible proteins, and a suite of conserved TFs that are involved in stress responses across the animal kingdom, including bZIP TFs ATF3, C/EBP (CHOP), CREB and MAF, HIFa, and NF-κB (Fig 3B–D) [44,45]. The most interconnected genes in this network are conserved heat shock and chaperone proteins, including HSP70 (HSPA), HSP40 (DNAJ), HSP83, HSP60, HSP20 (Protein lethal(2)essential for life), HSP52/27/28 (HSPB1) and HSP10; chaperonin containing TCP1 subunits 2, 4, and 8 (CCT2, 4 and 8); and BAG family molecular chaperone regulator 4. Notably, there is no evidence that the genes in this module, many of which are transiently expressed in response to acute stresses in other animals [45–48], were activated because of a marked change in environmental conditions, such as a marine heat wave or hypoxia on the source reefs (Fig 1A and 1B). Gravid males collected from the same reef locations and at the same time did not show up-regulation of these stress genes. Also significant is that these stress-related genes are up-regulated in female skin in the summer (Fig 3D), in contrast to other protein-coding genes in this tissue that are constitutively expressed or down-regulated in the summer (S1 Fig and S5 Table). Together, these findings lead us to conclude that the activation of stress proteins in gravid females reflects an organism-wide condition that may relate to the physiological taxing nature of being in a reproductive state.

## Discussion

By comparing protein-coding gene expression in 7 somatic tissues from wild COTS in the summer and winter, we have revealed new insights into seasonal changes in this coral reef pest. The procurement of RNA from small, targeted biopsies from multiple starfish immediately after their removal from the reef yielded consistent gene expression profiles within a given tissue, season, and sex, supporting the proposition that this replicated transcriptomic dataset accurately reflects natural gene activity [29]. This proposition is further evidenced by each tissue transcriptome being enriched in genes whose roles reflect the known biological function of that tissue. Thus, in addition to identifying differences between seasons and sexes, our approach provides new insights into the natural functioning of tissues and organs in the wild.

For example, compared to other COTS tissues, the radial nerve, sensory tentacles, and tube feet up-regulate neural and sensory genes, consistent with sensory roles of these organs in COTS and other echinoderms [28,30–34,36]. This includes the marked up-regulation of a diversity of chemosensory GCPRs in the most distal sensory tentacles and signalling neuropeptides in the radial nerve in the summer. By contrast, aboral papulae, skin and spines up-regulate genes involved in immunity and cilia function, consistent with skin having a general immune function and papulae being comprised of ciliated cells that create countercurrents involved in respiration [35,36]. Papulae additionally express diverse GPCRs, secreted proteins, TFs, and lineage-specific genes, suggesting this tissue has a wider sensory and regulatory role than previously appreciated. Coelomocytes isolated from wild COTS are probably derived from the perivisceral coelom, which has a diversity of functions in COTS and other echinoderms, including immunity and wound healing, nutrient transport, and waste removal [36,49–51]. We found that the coelomocyte transcriptomes are enriched in genes associated with microfilament function, consistent with phagocytes being a dominant cell type in this tissue as observed in other echinoderms [36,51].

All tissues express a unique repertoire of TFs, with neither oral nor aboral tissues uniquely sharing any TFs to the exclusion of all other tissues, despite their respective transcriptomes being more similar. Surprisingly, the tissues that have the highest diversity of differentially expressed TFs—the radial nerve and spines—also share the most TFs with each other. As these 2 tissues are functionally different [36], and express transcriptomes of similar complexity to the other tissues, it is unclear why they differentially express a greater diversity of TFs. The presence of a diverse TF repertoire in spines is consistent with this tissue playing a diversity of roles in conspecific communication that include both releasing and receiving external signals [21]. This suggestion is further supported by spines expressing the most lineage-specific genes of any tissue, many of which appear to be released into the sea water surrounding aggregating COTS [21].

Most of the 2,079 seasonally expressed protein-coding genes were significantly up-regulated in the summer (71.8%). A subset of these may play a role in conspecific communication underlying pre-spawning aggregations and synchronised spawning events [28], including 21 putative chemosensory GPCRs, 189 secreted proteins, and 285 lineage-specific genes, many of which are expressed in a sex-specific manner. This finding, combined with the differential activation and repression of other sex-specific genes in the summer, suggests that gene activity in the summer is maximally different between sexes, and is likely to underlie behaviours that ensure the pre-spawning formation of male and female aggregations in the wild.

Correlated with the up-regulation of conspecific communication and other protein-coding genes in the summer are 89 TFs; in contrast, only 13 TFs are up-regulated in the winter. Although most summer TFs are expressed in specific somatic tissues and are thus likely to be part of specific cell and tissue type gene regulatory networks, a subset is co-expressed across all tissues in the summer. These widely expressed TFs may be coordinating the complex tissue-specific transcriptional changes that occur as seawater temperature and day length increases, and as COTS reproductively mature. Among these co-expressed TFs in the summer are 6 nuclear receptors that are known in vertebrates to be influenced by seasonal hormones [43,52]. Putative nuclear receptor ligands, such as steroids, have been detected in COTS and in other echinoderms, and may be playing a seasonal role in reproduction [51,53–55]. These findings suggest there are conserved deuterostome—and maybe even bilaterian or metazoan—mechanisms to regulate circannual physiological changes via the activation of nuclear receptors.

Unexpectedly, we observed many highly conserved, stress-related chaperones and TFs up-regulated in somatic tissues of gravid (summer) females, but not males. A single female COTS

can produce more than 100 million eggs in one spawning season and the ovary can comprise up to 34% of adult body mass, reflecting the very large energy investment in reproduction [7]. On the reef, female COTS are most fecund when sea temperature is near maximal and oxygen levels are lower and have greatest diel fluctuations [56]. As there was no evidence of abnormally high temperatures or other environmental stressors at the time the COTS were sampled, we infer that the up-regulation of stress proteins may reflect the overall physiological state of female COTS that are channeling a large proportion of their nutrients to oocytes. It would be interesting to see if this phenomenon occurs in other highly fecund echinoderms and marine invertebrates, as it is currently unclear if this is a conserved phenomenon used in marine animals that seasonally deploy a large proportion of their resources to the production of gametes.

In conclusion, our atlas of somatic tissue transcriptomes from wild COTS in the summer and winter enables the identification of molecular factors that underpin season- and sex-specific biological processes. Given the large-scale transcriptional changes that occur when COTS are translocated from the wild [29], it appears unlikely that the same depth of insight would be achieved by sampling captive starfish. The seasonal changes in gene expression have revealed a complex, and often sex-specific, interplay between regulatory, signalling, and structural genes. Notably, this includes tissue-specific expression profiles that overlap substantially with adjacent or related tissues, and the large-scale activation of genes in the summer, many of which are differentially up-regulated in one sex and are involved in cell signalling and conspecific communication. This analysis also uncovered a cryptic physiological state in gravid females that includes a systemic up-regulation of conserved stress response genes that may have a protective role prior to spawning. Beyond its utility in understanding the biology of this coral reef pest and providing molecular leads for its future biocontrol, this atlas provides a means to understand circannual rhythms in tropical and coral reef animals where seasonal differences are less pronounced than elsewhere on Earth.

## Materials and methods

### Sampling of wild-caught crown-of-thorns starfish and CEL-Seq2 library construction

Adult COTS (*Acanthaster* cf. *solaris*; member of the *A.* "*planci*" species complex) [1–3] were collected from Davies and Lynch's Reef on the Great Barrier Reef (Fig 1A), and tissue RNAs were isolated as previously described [28]. Care was taken to ensure no overlap between tissues (e.g., skin had no papulae and vice versa). Briefly, COTS were collected by hand from the reef and housed in ambient seawater onboard the vessel. Tissues were biopsied and transferred into RNALater less than 2 h after the starfish were removed from the reef. Seven unsexed individuals were collected during the winter (August 2019), and 7 and 6 gravid females and males, respectively, were collected during the summer reproductive season (December 2019). Seven somatic tissues—coelomocytes, sensory tentacles, tube feet, radial nerves, skin, papulae, and spines—were isolated from each female, male and winter individual, except coelomocytes and spines were not collected from 2 and 1 individuals in the winter, respectively.

RNA isolation and quality-assessment, and CEL-Seq2 library construction all were performed as previously described [28]. Tissues from each individual were kept separate throughout the procedures, and each individual sample was uniquely barcoded prior to pooling and sequencing [57]. The CEL-Seq2 libraries were sequenced on an Illumina HiSeq X ten platform at NovogeneAIT Genomics in Singapore.

Raw reads were assessed for quality and adaptor contents using FastQC [58] and analysed using the CEL-Seq2 pipeline (https://github.com/yanailab/CEL-Seq-pipeline; version 1.0) [29]. Reads were trimmed to 35 bp, demultiplexed and mapped to the GBR v1.1 genome using

Bowtie2 [59]. Transcript counts of each protein coding sequence were generated using HTSeq [60]. Samples with mapped transcripts <0.5 million were discarded as low-quality [61]. Raw read counts and normalized and average transcripts per million (TPM) for all transcriptomes can be found at https://doi.org/10.5281/zenodo.10831187.

## Identification of lineage-specific classes of expressed genes

TFs and chaperones were identified based on their Blast2GO annotation [29]. GPCRs were identified by gene classification established by Hall and colleagues [21]. SignalP 5.0 was used to determine if COTS GBR v1.1 gene models [29] possessed signal peptides indicating that they potentially encoded for secreted proteins [62]. These gene models were then assessed for transmembrane domains that would indicate they encoded membrane-bound rather than secreted proteins using TMHMM Server v. 2.0 [63,64]. Proteins were considered secreted if they possessed a predicted signal peptide but no transmembrane domain. Identified secreted proteins were then classified into 6 categories: hydrolytic enzymes; other enzymes; enzyme inhibitors; structural/signalling proteins; conserved uncharacterised proteins; and novel uncharacterised proteins [21].

We used the reference genomes of 5 echinoderms, downloaded from NCBI (*Strongylocentrotus purpuratus* and *Apostichopus japonicus*) [65,66], Echinobase (*Asterias rubens* and *Patiria miniata*) [67], and from https://marinegenomics.oist.jp/cots/viewer/info?project_id=46 (COTS) [21], to identify lineage-specific genes. Orthofinder was used with default settings [68] to identify orthologous groups, gene duplication events, and putative species-specific coding sequences. COTS protein-coding genes that did not appear to share orthologues with other echinoderms or that had duplicates (paralogues) were deemed to be lineage-specific. A Venn diagram of the number of shared and lineage-specific orthologous groups of sequences was generated using https://bioinformatics.psb.ugent.be/webtools/Venn/. A species tree was generated using Orthofinder [68].

## Differential gene expression analyses

To determine potential read count errors associated with lowly expressed genes [69], we tested different expression threshold levels of an average of ≥0.25 and ≥1 read per gene per library for all replicates of a given tissue and opted for the expression threshold of ≥0.25 mean reads per tissue to increase the chance of detecting lowly expressed protein-coding genes (S4 Fig) [28]. Using Wald Test statistics in the DESeq2 package (v1.28.1) [70], gene expression levels with an adjusted *p*-value <0.05 between conditions (tissues, season, and sex) were deemed to be significantly differentially expressed. We conducted pairwise DESeq2 analyses between all tissues, and protein-coding genes that were consistently up-regulated in any given tissue were designated as tissue-specific. Similarly, for each tissue, the summer male and female groups were individually tested against the winter individuals using DESeq2. Protein-coding genes that were up-regulated in both the male and female group compared to winter were designated as up-regulated in summer and vice versa.

To detect modules of co-expressed genes from the transcriptome data, WGCNA was applied to normalised variance-stabilising transformed (VST) data [40]. Protein-coding genes expressed at low levels and with low expression variance across the libraries were filtered out. The remaining 9,159 genes were used in the WGCNA analysis. A signed network was constructed in WGCNA with specific parameter settings of power = 7, TOMType = "signed Nowick" and minModuleSize = 100 [40]. Networks significantly correlated with the summer sampling were visualised in Cytoscape (v. 3.9.1) [71] and were filtered based on edge weight.

All DESeq2 pairwise test statistics and DESeq2 and WGCNA R scripts can be found at https://doi.org/10.5281/zenodo.10831187.

### Enrichment analyses and data visualisation

Gene Ontology (GO) enrichment analyses of protein-coding genes were performed using Blast2GO as previously described [29], using the Fisher's exact test function available on the "clusterProfiler" package (v4.2.2) [72]. GO terms with a false discovery rate (FDR) adjusted $p$-value of <0.05 were considered enriched. GO enrichment analysis for the lineage-specific genes was performed against all the genes in the COTS GBR v1.1 genome [29], whereas GO enrichment analyses of tissue-specific or seasonally up-regulated genes were performed against the genes expressed in specific tissues. KOBAS-i was used to test the statistical enrichment of KEGG pathways in differentially expressed protein-coding genes, with a FDR-corrected $p$-value of <0.05 [73]. Genes expressed in each tissue were used as reference for the enrichment analyses. GO and KEGG enrichment analyses were visualised using the "ggplot2" R package (v3.3.6) [74]. GO enrichment scripts can be found at https://doi.org/10.5281/zenodo.10831187.

PCAs were used to visualise differences in gene expression between tissues and seasons. PCAs were performed on VST counts obtained with DESeq2. Heatmaps were created for each tissue separately, to visualise expression patterns of genes of interest, using TPM normalised raw reads, in the R package "pheatmap" [75,76].

All analyses and visualisations were performed in RStudio version 4.0.2 [77].

### Supporting information

**S1 Fig. Tissue, season, and sex gene expression.** **(A)** Hierarchical clustered heatmap of Pearson correlation of expressed protein-coding genes across the 7 tissue transcriptomes. **(B)** Heatmap of the 2,079 protein-coding genes that are significantly differentially expressed between seasons in at least 1 tissue (adjusted $p$-value <0.05). The genes up-regulated in the summer (yellow) and winter (blue) are at the top and bottom of the heatmap, respectively. The "All" heatmap shows the average expression across all tissues across for both sexes and seasons. **(C)** Heatmap of the TFs that are significantly differentially expressed COTS tissues (adjusted $p$-value <0.05). The data underlying this figure can be found in S3 and S5 Tables and at https://doi.org/10.5281/zenodo.10831187.
(PDF)

**S2 Fig. Identification and expression of COTS-specific genes.** **(A)** Venn diagram showing the overlap of orthologous gene groups among 5 echinoderms, the sea urchin *Strongylocentrotus purpuratus* [65], the sea cucumber *Apostichopus japonicus* [66], and starfish *Asterias rubens*, *Patiria miniata* [67], and *Acanthaster* cf. *solaris* (COTS) [21,29]. COTS has 506 unique gene groups. **(B)** Phylogenetic tree of the 5 echinoderms showing gene duplication events at each node and branch. The taxonomic ranks are listed on the right. The numbers next to each species name are the total number of lineage-specific gene duplication events as detected by Orthofinder. **(C)** GO term enrichments of COTS-specific genes. The x-axis shows the GO terms and the y-axis shows the number of protein-coding genes in each term. **(D, E)** Expression profiles of lineage-specific GPCRs and secreted proteins in wild COTS. The heatmaps show the scaled expression levels (z-score) based on TPM normalized reads for each tissue in 7 females, 6 males, and 7 winter individuals. The "All" heatmap shows the average expression across all groups and tissues. The heatmaps are colour-coded (right) by significant differential expression (DESeq2; adjusted $p$-value <0.05): not significantly differentially expressed (brown), significantly up-regulated in the summer (yellow) and significantly up-regulated in

the winter (blue). The data underlying this figure can be found in S7 Table and at https://doi. org/10.5281/zenodo.10831187.
(PDF)

**S3 Fig. WGCNA module identification and correlation analysis.** **(A)** Clustering dendrogram of all samples. The colour bar at the bottom corresponds to COTS tissues: green, radial nerve; pink, tube feet; purple, sensory tentacles; brown, papulae; turquoise, skin; blue, spines; red, coelomocytes. **(B)** Scale-free topology model fit for different soft-thresholding powers (left). The signed $R^2$ value measures how well the scale-free topology assumption fits the observed network connectivity distribution. Mean network connectivity for different soft-thresholding powers (right). The mean connectivity is the average number of connections (edges) per node (gene) in the network for a given power value. **(C)** Module-trait associations. Each column represents a trait (tissue/season/sex), and each row represents an identified module. Red and blue colour notes positive and negative correlation with the trait, respectively. Numbers in each cell represent the correlation coefficient and the *p*-value (in brackets). **(D)** Gene interaction networks for the turquoise module (edge weights >0.164). The size of each coding sequence (node) in the network corresponds to the significance of differential expression (up-regulated in the summer; *p*-adjusted value in DESeq2). The colour corresponds to the log2fold change in gene expression: red, up-regulated in summer; blue, up-regulated in winter. **(E)** Expression profiles of the core hub genes in the turquoise module. This heatmap shows the scaled expression levels (z-score) based on TPM normalised reads for each tissue in females, males, and winter individuals. The "All" heatmap shows the average expression across all groups and tissues. The data underlying this figure can be found in S8 Table and at https://doi. org/10.5281/zenodo.10831187.
(PDF)

**S4 Fig. Comparison of expression threshold of seasonal transcriptomes.** PCA of tissue transcriptomes with an average expression threshold $\geq 0.25$ **(A)** and $\geq 1$ **(B)**; 95% confidence ellipses shown. Pink, tube feet; purple, sensory tentacles; green, radial nerve; turquoise, skin; brown, papulae; blue, spines; red, coelomocytes.
(PDF)

**S1 Table. Mapping rates of CEL-Seq2 transcriptome reads to the COTS GBR v1.1 genome using Bowtie2.**
(XLSX)

**S2 Table. Summary of expression profiles of protein-coding genes in different COTS tissues.** The table shows the presence (1) or absence (0) of reads for each gene in 7 tissues: coelomocytes (Co), skin (Sk), spines (Sp), papulae (Pa), radial nerve (RN), tube feet (TF), and sensory tentacles (ST). The table also provides information on the function, annotation, and classification of each protein-coding gene based on various databases and sources.
(XLSX)

**S3 Table. Summary of significantly up-regulated protein-coding genes in each tissue. S3.1 Table.** Coelomocytes. **S3.2 Table.** Papulae. **S3.3 Table.** Radial nerve. **S3.4 Table.** Skin. **S3.5 Table**. Spines. **S3.6 Table**. Sensory tentacles. **S3.7 Table.** Tube feet. **S3.8 Table.** Gene ontology (GO) enrichments based on significantly up-regulated genes in each tissue (adjusted *p*-value <0.05).
(XLSX)

**S4 Table. Summary of significantly up-regulated protein-coding genes in oral and aboral tissues. S4.1 Table.** List of significantly up-regulated (1) and not significantly up-regulated (0)

genes in oral and aboral tissues (adjusted *p*-value <0.05). **S4.2 Table.** Significant gene ontology (GO) enrichments based on up-regulated genes in oral and aboral tissues.
(XLSX)

**S5 Table. Summary of significantly differentially expressed protein-coding genes between summer and winter. S5.1 Table.** DESeq2 was applied pairwise comparing summer and winter expression for each tissue. Significant up-regulation (1) and no significant up-regulation (0) was depicted across all tissues (adjusted *p*-value <0.05). **S5.2 Table.** Gene ontology (GO) enrichments based on significantly up-regulated genes in summer and winter for all tissues. **S5.3 Table.** KEGG enrichments based on significantly up-regulated genes in summer and winter for all tissues. **S5.4 Table.** TFs significantly up-regulated in the summer. **S5.5 Table.** TFs significantly up-regulated in the winter.
(XLSX)

**S6 Table. Tissue expression of 283 genes significantly differentially expressed between gravid female and male COTS [28].** Protein-coding genes are listed in the same order as in Fig 2D. **S6.1 Table.** Protein-coding genes significantly up-regulated in gravid females compared to gravid males. **S6.2 Table.** Average tissue expression levels of genes significantly up-regulated in gravid females compared to gravid males. **S6.3 Table.** Protein-coding genes significantly up-regulated in gravid males compared to gravid females. **S6.4 Table**. Average tissue expression levels of genes significantly up-regulated in gravid males compared to gravid females.
(XLSX)

**S7 Table. Lineage-specific protein-coding genes that are differentially expressed in COTS somatic tissues. S7.1 Table.** Overall results and statistics output from Orthofinder. **S7.2 Table.** Orthofinder statistics for the sea urchin *S. purpuratus*; the sea cucumber *A. japonicus*; and starfish *A. rubens*, *P. miniata*, and COTS. **S7.3 Table.** Orthogroups present in the echinoderm genomes. **S7.4 Table.** Gene duplications present in the echinoderm genomes. **S7.5 Table.** Number of differentially expressed lineage-specific protein-coding genes expressed in each COTS tissue. **S7.6 Table.** All significantly differentially expressed lineage-specific genes, across both season and sex. Samples were compared pairwise, female summer (F) vs. male summer (M); F summer vs. winter; and M summer vs. winter, where 1 depicts significant up-regulation. **S7.7 Table.** Significant GO enrichments based on differentially expressed lineage-specific protein-coding genes.
(XLSX)

**S8 Table. Co-expressed protein-coding genes in cyan, pink, and turquoise co-expression modules. S8.1 Table.** WGCNA output data showing module membership (MM) of each gene. A higher absolute value indicates a stronger correlation with the module. **S8.2 Table.** WGCNA output data showing gene trait significance (GS), which is a measure of how closely a gene is related to the trait(s) of interest. A higher absolute value indicates a stronger correlation with the trait. **S8.3 Table.** Co-expressed protein-coding genes in the cyan module. **S8.4 Table.** Significant GO enrichments based on protein-coding genes in the cyan module. **S8.5 Table.** Co-expressed coding sequence in the pink module. **S8.6 Table.** Significant GO enrichments based on protein-coding genes in the pink module. **S8.7 Table.** Co-expressed protein-coding genes in the turquoise module. **S8.8 Table.** Significant GO enrichments based on genes in the turquoise module.
(XLSX)

## Acknowledgments

We thank the Great Barrier Reef Foundation and Association of Marine Park Tourism Operators Limited (AMPTO) for their financial and logistical support; the crew and divers of the AMPTO vessels that assisted in the field collection of COTS; and Nick Rhodes, Michael Thang, Gareth Price, and Dominique Gorse from the Queensland Cyber Infrastructure Foundation for development of the COTS genome browser and computing assistance.

## Author Contributions

**Conceptualization:** Marie Morin, Mathias Jönsson, Sandie M. Degnan, Bernard M. Degnan.

**Data curation:** Marie Morin, Mathias Jönsson.

**Formal analysis:** Marie Morin, Mathias Jönsson.

**Funding acquisition:** Conan K. Wang, David J. Craik, Sandie M. Degnan, Bernard M. Degnan.

**Investigation:** Marie Morin, Mathias Jönsson, Sandie M. Degnan, Bernard M. Degnan.

**Methodology:** Marie Morin, Mathias Jönsson, Sandie M. Degnan, Bernard M. Degnan.

**Project administration:** Sandie M. Degnan, Bernard M. Degnan.

**Resources:** Marie Morin, Mathias Jönsson, Sandie M. Degnan, Bernard M. Degnan.

**Supervision:** Sandie M. Degnan, Bernard M. Degnan.

**Validation:** Marie Morin, Mathias Jönsson, Sandie M. Degnan, Bernard M. Degnan.

**Visualization:** Marie Morin, Mathias Jönsson, Sandie M. Degnan, Bernard M. Degnan.

**Writing – original draft:** Marie Morin, Mathias Jönsson, Sandie M. Degnan, Bernard M. Degnan.

**Writing – review & editing:** Marie Morin, Mathias Jönsson, Conan K. Wang, David J. Craik, Sandie M. Degnan, Bernard M. Degnan.

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
