## [Editor Report · Decision Letter 0]

4 Jan 2024

Dear Bernie, 

Thank you for submitting your manuscript entitled "Seasonal tissue-specific gene expression reveals reproductive and stress-related transcriptional systems in wild crown-of-thorns starfish" for consideration as a Methods and Resources by PLOS Biology. Many thanks also for your patience over the holiday period.

Your manuscript has now been evaluated by the PLOS Biology editorial staff, as well as by an academic editor with relevant expertise, and I'm writing to let you know that we would like to send your submission out for external peer review.

Once your full submission is complete, your paper will undergo a series of checks in preparation for peer review. After your manuscript has passed the checks it will be sent out for review. To provide the metadata for your submission, please Login to Editorial Manager (https://www.editorialmanager.com/pbiology) within two working days, i.e. by Jan 08 2024 11:59PM.

Kind regards,

Roli

Roland Roberts, PhD

Senior Editor

PLOS Biology

rroberts@plos.org

---

## [Decision Letter · Decision Letter 1]

8 Mar 2024

Dear Bernie,

Thank you for your patience while your manuscript "Seasonal tissue-specific gene expression reveals reproductive and stress-related transcriptional systems in wild crown-of-thorns starfish" went through peer-review at PLOS Biology. Your manuscript has now been evaluated by the PLOS Biology editors, an Academic Editor with relevant expertise, and by three independent reviewers.

Reviewer #1 is positive, having no actual requests, but regretting the lack of gonadal data and validation by qPCR and/or FISH (we will not require these). Reviewer #2 is also positive, but has constructive suggestions to improve the contextualisation in the Intro and the dataviz (converting Table 2 into a heatmap; reformatting the heatmaps in Figs 2 and 3), and wonders if you could help readers interpret the WGCNA results more easily. Reviewer #3 is positive and has very minor requests.

In light of the reviews, which you will find at the end of this email, we are pleased to offer you the opportunity to address the comments from the reviewers in a revision that we anticipate should not take you very long. We will then assess your revised manuscript and your response to the reviewers' comments with our Academic Editor aiming to avoid further rounds of peer-review, although might need to consult with the reviewers, depending on the nature of the revisions.

IMPORTANT:

a) Please address the concerns raised by the reviewers; in particular, we think that the presentational suggestions from reviewer #2 will help the accessibility and appeal of your paper. 

b) Please comply with our Data Policy; specifically, we need you to supply the numerical values underlying Figs 1BCDE, 2ABCDE, 3ABCDEFG, S1CDE, S2ABC, S3AB, either as a supplementary data file or as a permanent DOI’d deposition.

c) Please cite the location of these data clearly in all relevant main and supplementary Figure legends, e.g. “The data underlying this Figure can be found in S1 Data” or “The data underlying this Figure can be found in https://doi.org/10.5281/zenodo.XXXXX”

d) Please make any custom code available, either as a supplementary file or as part of your data deposition.

**IMPORTANT - SUBMITTING YOUR REVISION**

*Resubmission Checklist*

*Published Peer Review*

*PLOS Data Policy*

Sincerely,

Roli

Roland Roberts, PhD

Senior Editor

PLOS Biology

rroberts@plos.org

REVIEWERS' COMMENTS:

Reviewer #1:

PBIOLOGY-D-23-03373R1

Seasonal tissue-specific gene expression reveals reproductive and stress-related transcriptional systems in wild crown-of-thorns starfish

Summary 

Wildtype animals in their native habitats, are often overlooked when considering experimental models. Most animals though, do not live in laboratories, housed under constant conditions, with ample food, conditioned light, and clean conditions. Biology is impacted by changes in the world, by reproduction, temperature change and physical conditions that constantly vary. The current dataset is a very nice demonstration of such dynamic changes in an important animal model. 

The data are RNA-seq experiments of different tissues from different animals from different times of the year. As such - it is exactly what investigators need in order to understand biology. These datasets will be very useful in looking at any further analysis of especially marine organisms. 

Critique

It is unfortunate that the investigators did not include gonads in their analysis. Although gonads have been sequenced from this animal, they would serve as an excellent and essential foundation for measuring changes in other somatic tissues. Without the gonads, we are missing a key baseline for which all other tissues can be compared. It would serve as a seasonal control, especially since in this animal, the reproductive cycles are so well known. Further, the gonads may well be the regulators of all the other tissues sampled and we are left guessing what correlative changes we might find were those data also present. 

Further, as valuable as the datasets are - it would be important to have some supporting data to test RNA-values by e.g. qPCR, and to have in situ RNA hybridizations to see which cells of the various tissues the genes are actually expressed. These correlative experiments would help the reader better understand context for within the space that the data must sit. 

Overall conclusion

Valuable datasets are given and will be useful by the wider community. The impact will however be limited by the absence of key foundational tissue measurements and follow-up examples. 

Reviewer #2:

In this manuscript, the authors thoroughly analyze tissue transcriptomes from the crown-of-thorns starfish (COTS) across the summer and winter seasons. Investigating seven major tissue types revealed seasonal and sex-specific variations in gene expression. Despite its complexity, the employment of weighted gene correlation network analysis (WGCNA) for identifying co-expressed genes effectively highlights the summer upregulation of various signaling and transcription factors, mainly in females. Nevertheless, while valuable, the extensive tissue data appears to have detracted from a clear presentation of seasonal differences. Although the manuscript is competently written and features adequate visual representation, refinements in phrasing and data visualization could significantly improve its publication readiness and accessibility to a broader audience.

Major comments:

1. The introduction could be expanded to give readers a better understanding of the topic, especially regarding seasonal expression changes in other animals.

2. Presenting tissue-specific seasonal changes is currently cumbersome; for instance, Table 2 is challenging to interpret and offers limited experimental insight. Converting this into a heatmap integrated into Fig. 2 might enhance readability.

3. Patterns like those in Fig. 2A and B are more discernible. Reformating heatmaps in Figs. 2 and 3 could make seasonal and sex-specific expression patterns more apparent, highlighting areas of co-expressed genes with significant upregulation or downregulation.

4. While using WGCNA to identify co-expressed genes is logical, interpreting these results is complex. Could statistical analysis pinpoint season-specific genes and categorize them by tissue specificity? Alternatively, focusing only on seasonal and sex differences in Fig. 3A might simplify the analysis.

5. The manuscript uses interchangeable terms for 'protein-coding gene,' such as protein-coding mRNAs, coding sequences, and protein-coding sequences. A uniform terminology is recommended.

Minor comments:

6. "Like most marine animals, ..." could be more precisely directed at marine invertebrates to avoid confusion with marine mammals.

7. The color coding for cyan and turquoise modules is too similar, making them hard to differentiate.

8. Adding line numbers would facilitate more accessible manuscript review.

Reviewer #3:

This work by Morin et al. provides interesting datasets about the physiological ecology of the wild crown-of-thorns starfish by the tissue-, season- and sex-level broad transcriptomic analysis. Although the method of this study seems similar to those of previous research including previous works of the authors themselves, the Cell-Seq2 transcriptomic datasets of the COTS caught in different seasons will be the good references to study about the annual reproductive cycle, and other year-round behaviors and ecologies of the echinoderms. Handling of the datasets in this study seems good, and the discussion for each result is mostly reasonable. Following are some minor points to be collected and/or explained.

Materials and Methods

* Section 2, Paragraph 2: Is the reference genome of Apostichopus japonicus downloaded by NCBI? If it is true, please adjust the place of parentheses to avoid ambiguity.

* Section 3, Paragraph 2: Please describe what does "VST" mean in this paragraph instead of S4, P2.

Fig 1A:

* Please describe the value of the scale bar.

S1 Fig. B: 

* What the numbers on the right of each species' name mean?

* It would be better to refer how this result (gene duplication events at each node and branch) is related to the authors' discussion in the main manuscript.

---

## [Editor Report · Decision Letter 2]

10 Apr 2024

Dear Bernie,

Thank you for the submission of your revised Methods and Resources "Seasonal tissue-specific gene expression in wild crown-of-thorns starfish reveals reproductive and stress-related transcriptional systems" for publication in PLOS Biology. On behalf of my colleagues and the Academic Editor, Yi-Hsien Su, I'm pleased to say that we can in principle accept your manuscript for publication, provided you address any remaining formatting and reporting issues. These will be detailed in an email you should receive within 2-3 business days from our colleagues in the journal operations team; no action is required from you until then. Please note that we will not be able to formally accept your manuscript and schedule it for publication until you have completed any requested changes.

IMPORTANT:

a) I've taken the liberty of tweaking your Title slightly, bringing the notorious study organism further up the text for extra prominence.

b) Thanks for providing the code in Github. However, because Github depositions can be readily changed or deleted, please make a permanent DOI’d copy (e.g. in Zenodo) and provide this URL in the manuscript? I've asked my colleagues to include this request in their list.

Sincerely, 

Roli

Senior Editor

PLOS Biology

rroberts@plos.org